# What Do We Know About *Cryptococcus* spp. in Portugal? One Health Systematic Review in a Comprehensive 13-Year Retrospective Study (2013–2025)

**DOI:** 10.3390/jof11090672

**Published:** 2025-09-12

**Authors:** Ricardo Lopes, Andreia Garcês, Hugo Lima de Carvalho, Vanessa Silva, Filipe Sampaio, Cátia Fernandes, Gonçalo Barros, Alexandre Sardinha de Brito, Ana Rita Silva, Elsa Leclerc Duarte, Luís Cardoso, Ana Cláudia Coelho

**Affiliations:** 1Department of Veterinary Sciences, University of Trás-os-Montes e Alto Douro (UTAD), 5000-801 Vila Real, Portugal; lcardoso@utad.pt (L.C.); accoelho@utad.pt (A.C.C.); 2Department of Veterinary and Animal Sciences, University Institute of Health Sciences (IUCS), CESPU, 4585-116 Gandra, Portugal; 3CEDIVET Veterinary Laboratories, Lionesa Business Hub, R. Lionesa 446 C24, 4465-671 Leça do Balio, Portugal; hugo.carvalho@cedivet.pt (H.L.d.C.); vanessa.campos.silva@cedivet.pt (V.S.); filipe.sampaio@cedivet.pt (F.S.); goncalo.barros@cedivet.pt (G.B.); 4Wildlife Rehabilitation Centre (CRAS), Veterinary Teaching Hospital, University of Trás-os-Montes e Alto Douro (UTAD), 5000-801 Vila Real, Portugal; andreiamvg@gmail.com; 5Animal and Veterinary Research Centre (CECAV), Associate Laboratory for Animal and Veterinary Sciences (AL4AnimalS), University of Trás-os-Montes e Alto Douro (UTAD), 5000-801 Vila Real, Portugal; 6Cytology and Hematology Diagnostic Services, Laboratory of Histology and Embryology, Department of Microscopy, ICBAS-School of Medicine and Biomedical Sciences, University of Porto (U. Porto), Rua de Jorge Viterbo Ferreira, 228, 4050-313 Porto, Portugal; 7AniCura Santa Marinha Veterinary Hospital, R. Dom Henrique de Cernache 183, 4400-625 Vila Nova de Gaia, Portugal; catia.fernandes@anicura.pt; 8Independent Researcher, 70378 Stuttgart, Baden-Württemberg, Germany; alexandresardbrito@gmail.com; 9Molecular Diagnostics Laboratory, Unilabs Portugal, Lionesa Business Hub, R. Lionesa 446 C24, 4465-671 Leça do Balio, Portugal; ana.rita.silva@unilabs.com; 10Department of Veterinary Medicine, School of Science and Technology, University of Évora, Polo da Mitra, Apartado 94, 7002-554 Évora, Portugal; 11Mediterranean Institute for Agriculture, Environment and Development (MED), Global Change and Sustainability Institute (CHANGE), University of Évora, Polo da Mitra, Apartado 94, 7002-554 Évora, Portugal

**Keywords:** cryptococcosis, *Cryptococcus gattii* complex, *Cryptococcus neoformans* complex, environmental reservoirs, MALDI-TOF MS, One Health, Planetary Health, Portugal, systematic review, zoonoses

## Abstract

Cryptococcosis, caused by the *Cryptococcus neoformans* and *Cryptococcus gattii* species complexes (pathogenic *Cryptococcus* spp.), is an environmentally acquired mycosis of One Health relevance. This study integrates a PRISMA-compliant systematic review (2000–2025) of Portuguese animal, human, and environmental reports with a 13-year retrospective dataset of laboratory-confirmed veterinary cryptococcosis cases (2013–2025). Clinical specimens were cultured and identified by MALDI-TOF mass spectrometry, and associations were assessed using χ^2^ and Fisher’s exact tests. Of 1059 submissions, 48 (4.5%) were culture-positive: 6.8% of canine, 5.3% of feline samples, and 4.0% of avian samples, with no detections in other vertebrate groups (*p* = 0.705). *Cryptococcus neoformans* predominated in carnivores (73.7%), while *Papiliotrema laurentii* (formerly *Cryptococcus laurentii)* was most frequent in birds (86.2%). Infection was not associated with sex or age. Seasonality was evident, with a July peak and summer predominance (*p* = 0.010). Most cases were from the Centre region (62.5%), with significant regional variation of *Cryptococcus* spp. distribution (*p* < 0.001). The systematic review confirmed autochthonous *C. gattii* complex disease and widespread *C. neoformans* contamination in pigeon guano and arboreal niches. These findings demonstrate a compartmentalised eco-epidemiology, reinforcing the need for integrated molecular typing, antifungal susceptibility testing, and coordinated human–animal–environment surveillance to inform targeted prevention and control strategies in Portugal.

## 1. Introduction

Cryptococcosis is a systemic mycosis of global distribution, primarily caused by members of the *Cryptococcus neoformans* and *Cryptococcus gattii* species complexes (pathogenic *Cryptococcus* spp.) [1,2]. These encapsulated yeasts are opportunistic pathogens capable of infecting a wide range of hosts, including humans, companion animals, livestock, and wildlife [2,3]. Although the disease is best known as an important opportunistic infection in immunocompromised people [2,4], its veterinary importance has gained increasing attention in recent decades [2,5]. Domestic animals not only suffer direct morbidity and mortality from cryptococcal disease but may also act as sentinels of environmental exposure, thereby contributing to One Health and Planetary Health-oriented surveillance of fungal pathogens [6]. Although historically considered uncommon in Europe relative to endemic regions such as Australia or parts of the Americas, the last few decades have seen an increase in reported veterinary cases and environmental isolations across multiple European countries [7].

The taxonomy of pathogenic *Cryptococcus* spp. has undergone considerable revision, and the complexes now comprehend multiple species with distinct ecological niches, epidemiological patterns, and virulence traits. The most recent revisions, supported by multilocus sequence typing and phylogenomic analyses, have reassigned several *Cryptococcus* sensu lato taxa to other genera within the order *Tremellales*. These include the conversion of *Cryptococcus laurentii* into *Papiliotrema laurentii*, *Cryptococcus humicola* into *Naganishia humicola*, *Cryptococcus neoformans* var. *uniguttulatus* into *Filobasidium uniguttulatum*, and *C. terreus* into *Solicoccozyma terreus* [8,9,10,11,12,13].

*C. neoformans* is most often associated with avian faecal-contaminated environments, especially pigeon droppings [14], whereas *C. gattii* is traditionally linked to certain tree species, such as eucalyptus, and can infect otherwise healthy hosts [12]. Both complexes, however, exhibit a wide geographic distribution and can be isolated from diverse environmental reservoirs. The infectious propagules, typically desiccated yeast cells or basidiospores, are acquired predominantly through inhalation, leading to a primary pulmonary infection that may remain localised or disseminate haematogenously to the central nervous system (CNS), eyes, skin, and other organs [15] (Figure 1).

In domestic animals, cryptococcosis has been documented most frequently in cats, dogs, and goats, but cases have also been reported in horses, ferrets, cattle, and various avian species [5,6,16,17]. The clinical manifestations vary considerably between host species and individuals. In cats, upper respiratory tract involvement, ocular lesions, and cutaneous nodules are common. In dogs, CNS disease is particularly prevalent. Goats may present with neurological signs, pneumonia, or mastitis, while birds can develop lesions in the beak or upper airways [5,18,19,20]. Such variation reflects differences in host susceptibility, immune status, and the infecting *Cryptococcus* spp. genotype [6,21].

From a public health perspective, the study of cryptococcosis in animals is important for several reasons. First, although *Cryptococcus* spp. are not directly transmitted between animals and humans in most circumstances, infected animals indicate the presence of environmental sources that may also pose risks to people. Second, veterinary cases can help map the distribution of specific molecular types, some of which exhibit different virulence profiles and antifungal susceptibilities [22]. Third, the clinical management of cryptococcosis in animals remains challenging, often requiring prolonged antifungal therapy, surgical intervention, or, in severe cases, euthanasia, thereby carrying economic and emotional costs [1].

Accurate diagnosis of cryptococcosis in animals requires a combination of clinical suspicion, cytological or histopathological visualisation of encapsulated yeasts, fungal culture, and increasingly, molecular identification [23]. Cryptococcal antigen detection in serum or cerebrospinal fluid can aid diagnosis, particularly in systemic cases, while genotyping allows for the differentiation of species and molecular types [24,25,26,27]. Such detailed characterisation is critical, as *C. gattii* infections, for instance, often occur in immunocompetent hosts and may require more aggressive or prolonged therapy compared to *C. neoformans* [16,23].

Diagnostic approaches in Europe mirror international guidelines: cytology and histopathology (visualising encapsulated yeasts), fungal culture, cryptococcal antigen testing (serum/cerebrospinal fluid—CSF), and molecular typing when available. The ABCD (European Advisory Board on Cat Diseases) guidance and veterinary reviews emphasise that antigen detection is sensitive for systemic disease, while culture and molecular tools add speciation and typing [28]. Treatment typically uses azoles (itraconazole and fluconazole), sometimes amphotericin B for severe disease or CNS involvement, and may require prolonged therapy and/or surgical resection of localised cryptococcomas. Prognosis varies by species, site of infection, and timeliness of therapy [5].

Despite some documented cases in Europe, there is currently no comprehensive review or systematic study addressing cryptococcosis in domestic animals in Portugal. Information remains scattered across isolated case reports and small-scale studies, with limited data on the prevalence, geographic distribution, and molecular epidemiology of the causative agents. Without consolidated knowledge, veterinarians may underestimate the likelihood of encountering cryptococcosis, potentially delaying diagnosis and appropriate treatment. Furthermore, the absence of coordinated surveillance hinders the identification of emerging trends, such as the appearance of novel molecular types or shifts in antifungal susceptibility patterns [29].

The present study aims to fill this knowledge gap by reviewing and analysing the occurrence of cryptococcosis in domestic animals in Portugal and Europe, drawing from published case reports, veterinary diagnostic records, and, where possible, environmental investigations. To the authors’ knowledge, this is the first systematic review to comprehensively characterise *Cryptococcus* spp. epidemiology in Portugal within a One Health and Planetary Health framework and the first to present a comprehensive 13-year retrospective study (2013–2025) of diagnosed cases of cryptococcosis.

## 2. Materials and Methods

### 2.1. Data Collection, Sampling, and Microbiological Analysis

All clinical submissions for mycological culture received by CEDIVET Veterinary Laboratories (Porto, Portugal) between 2013 and 2025 were considered for analysis, without implementation of active population-based screening. Samples originated from veterinary practices, including clinics and hospitals, across mainland Portugal and the Insular Autonomous Regions. Each submission included a standard laboratory requisition form, which provided clinical data for each animal, including species, breed, sex, age, relevant medical history, vaccination and prophylactic status, clinical suspicion or observed signs (e.g., nasal discharge, neurological deficits, and cutaneous lesions), and the diagnostic tests requested.

Samples were directly inoculated onto Sabouraud Dextrose Selective Agar, Emmons formulation, containing chloramphenicol and gentamicin (Thermo Scientific™, R01772, Waltham, MA, USA), incubated at 30 °C, and observed daily for the development of fungal colonies. Phenotypic identification of fungal isolates was based on both macroscopic and microscopic characteristics. Macroscopic evaluation included assessment of colony growth rate, topography, texture, surface and reverse pigmentation, and the presence and colour of any diffusible pigment. Microscopic identification was performed using Lactophenol Cotton Blue stain (Labkem^®^, Dublin, Ireland), evaluating hyphal features (colour, size, septation, and structure) and conidial characteristics (septation, shape, size, colour, wall texture, arrangement, and morphology and development of conidiogenous cells). When yeast-like colonies were obtained, typically smooth, glistening, and mucoid, without aerial mycelium, with colours ranging from cream to tan and occasionally producing brown pigment on specific media, microscopic examination focused on yeast morphology. This involved assessing cell size and shape, budding pattern (e.g., narrow-based budding in *Cryptococcus* spp.), and the possible presence of a polysaccharide capsule, inferred from a clear halo surrounding the cells (Figure 2). A representative cytological preparation from feline renal cryptococcoma showing encapsulated yeasts with narrow-based budding is provided in Figure 3.

For definitive identification, pure colonies were subjected to Matrix-Assisted Laser Desorption/Ionisation Time-of-Flight Mass Spectrometry (MALDI-TOF MS) using the microflex^®^ LT/SH system (Bruker Daltonics, Bremen, Germany). Protein extraction followed the manufacturer’s standard ethanol–formic acid protocol. Mass spectra were acquired and matched against the Bruker reference database, with identification scores interpreted according to the manufacturer’s criteria; scores above the validated threshold were considered reliable for species-level identification.

### 2.2. Statistical Analysis

All available data were extracted in digital format from the Sislab^®^ system (Glintt, Global Intelligent Technologies, Lisbon, Portugal) and transferred to Microsoft Excel^®^ (Microsoft Corp., Redmond, WA, USA) for preprocessing. Variables were categorised into geographical region (NUTS 2: Nomenclature of Territorial Units for Statistics), municipality, season, month, species, breed, sex, age (years), and microbiological result.

Statistical analysis was performed using JMP^®^ version 14.3 (SAS Institute Inc., Cary, NC, USA), DATAtab^®^ (DATAtab e.U., Graz, Austria), and MedCalc^®^ statistical software version 20.006 (MedCalc Software Ltd., Ostend, Belgium). Descriptive statistics summarised counts, proportions, and, where appropriate, medians with interquartile ranges (IQR). For species-level positivity, binomial proportions are given with two-sided 95% confidence intervals (CI) via the Wilson score method (no continuity correction). Associations between *Cryptococcus* positivity and categorical variables (species, breed, sex, age (years), geographical location, season, and month of sample collection) were assessed using the chi-squared (χ^2^) test, or Fisher’s exact test when expected cell frequencies were below five. Given age non-normality, groups were compared with the two-tailed Mann–Whitney U test. A *p*-value ≤ 0.05 was considered statistically significant [30].

### 2.3. Systematic Review on Cryptococcus spp. Infections in Portugal

The systematic review was conducted following the PRISMA (Preferred Reporting Items for Systematic Reviews and Meta-Analyses) guidelines. A comprehensive search retrieved peer-reviewed studies reporting detection, isolation, or positivity of *Cryptococcus* spp. in Portugal (2000–2025) across humans, animals (domestic, captive, and wild), and environmental samples (One Health/Planetary Health scopes); case reports/series without explicit denominators were synthesised narratively (no quantitative pooling where designs were incomparable).

The search was conducted in seven databases and institutional repositories: PubMed, Scopus, SciELO, RCAAP, INSA (Instituto Nacional de Saúde Doutor Ricardo Jorge), DGAV (Direção-Geral de Alimentação e Veterinária), and ECDC (European Centre for Disease Prevention and Control). The search terms used were “*Cryptococcus*” OR “*Cryptococcus neoformans*” OR “*Cryptococcus gattii*” OR “cryptococcosis” AND “Portugal”. Articles were screened for relevance, and only those with explicit Portuguese data on *Cryptococcus* infection or isolation, including species or molecular type identification, pathological/epidemiological findings, and clinical or environmental outcomes, were included in the final review. The final search was completed in August 2025.

## 3. Results

### 3.1. Microbiological Study of Cryptococcus spp. in Portugal Between 2013 and 2025

#### 3.1.1. General Positivity and Distribution by Host Species and Breed

Of the total of 1059 animals tested in this study, 48 (4.5%; 95% CI: 3.44–5.96%) were positive for *Cryptococcus* spp., while 1011 (95.5%; 95% CI: 94.04–96.56%) yielded negative results. The distribution of positive cases across animal species is detailed in Table 1. No positive results were recorded in equines (*n* = 9), lagomorphs (*n* = 7), rodents (*n* = 1), other mammals (*n* = 8; non-rodent, non-lagomorph species), or non-mammalian non-avian vertebrates (*n* = 11; chelonians, other reptiles, and fish).

Among the 29 positive avian samples, the materials collected included cloacal swabs (*n* = 24), egg contents (*n* = 2), organs (*n* = 1), and oropharyngeal swabs (*n* = 2). For the 13 canine samples, the materials collected included cerebrospinal fluid (*n* = 2), cutaneous swabs (*n* = 4), ear swabs (*n* = 1), urine (*n* = 1), nasal swabs (*n* = 4), and faecal material (*n* = 1). Among the six feline samples, the materials collected included nasal swabs (*n* = 2), cutaneous swabs (*n* = 3), and urine (*n* = 1).

Of the 48 *Cryptococcus* spp.-positive animals, 19 were canines and felines. Within these, *C. neoformans* was the most frequently identified species (*n* = 14; 73.7%), followed by *Papiliotrema laurentii* (formerly *C. laurentii*) (*n* = 3; 15.8%), *Naganishia humicola* (formerly *C. humicola*) (*n* = 1; 5.3%), and *Filobasidium uniguttulatum* (formerly *C. neoformans* var. *uniguttulatus*) (*n* = 1; 5.3%). The distribution of *Cryptococcus* spp. and taxonomically updated related species by breed in these two host groups is presented in Table 2.

Of the 48 *Cryptococcus* spp.-positive animals, the remaining 29 were avian species. Within these, *P. laurentii* (formerly *C. laurentii*) was the most frequently identified species (*n* = 25; 86.2%), followed by *C. neoformans* (*n* = 2; 6.9%), *F. uniguttulatum* (formerly *C. neoformans* var. *uniguttulatus*) (*n* = 1; 3.5%), and *Solicoccozyma terreus* (formerly *C. terreus*) (*n* = 1; 3.5%). The distribution of *Cryptococcus* spp. and taxonomically updated related species by avian species is presented in Table 3.

#### 3.1.2. Animal Species and Breeds

Regarding host species, the overall distribution is reported in Table 1. Overall *Cryptococcus* positivity was 4.5% (48/1059). Positivity by category was 5.3% in felines (6/113), 6.8% in canines (13/190), 4.0% in birds (29/720), and 0% in lagomorphs (0/7), rodents (0/1), equines (0/9), other mammals (0/8), and non-mammalian, non-avian vertebrates (0/11).

For inferential analysis, a chi-square test across the eight species categories found no evidence of association between host species and *Cryptococcus* positivity (χ^2^ = 4.63, *df* = 7, *p* = 0.705).

Among canine and feline samples, *Cryptococcus* positivity was concentrated in a limited number of breeds with very small denominators. Mixed-breed dogs (*n* = 7) and European/Domestic Shorthair cats (*n* = 4) accounted for the majority of detections, with additional positives recorded in Boston Terriers (*n* = 2) and single cases in German Shepherds, Saint Bernards, Yorkshire Terriers, French Bulldogs, Siberian Huskies, and Dobermanns (*n* = 1 each). Across all positives, the predominant agent was *Cryptococcus neoformans* (14/19; 73.7%), followed by *P. laurentii* (formerly *C. laurentii*) (3/19; 15.8%), *F. uniguttulatum* (formerly *C. neoformans* var. *uniguttulatus*) (1/19; 5.3%), and *N. humicola* (formerly *Cryptococcus humicola*) (1/19; 5.3%). Pearson’s chi-square test indicated a statistically significant association between breed and positivity (*p* < 0.001); however, given the very small sample sizes, with several breeds represented by a single submission yielding 100% positivity, these findings should be regarded as exploratory and do not provide robust evidence of a stable breed-level risk.

Across 720 avian submissions, *Cryptococcus* positivity totalled 29 cases (4.0%). Pearson’s chi-square test indicated a statistically significant association between avian taxon and positivity (χ^2^ = 694.76, *df* = 87, *p* < 0.001). Positives were concentrated in a small number of taxa, with Atlantic canary (*Serinus canaria*, *n* = 6), Emerald toucanet (*Aulacorhynchus prasinus*, *n* = 3), and Lovebird (*Agapornis* spp., *n* = 3) accounting for more than 40% of all detections. Other species were represented by one or two positive submissions each, including the African grey parrot, Blue-and-yellow macaw, Cockatiel, Eclectus parrot, European goldfinch, Parakeet, Red lory, Rock pigeon, Society finch, Stella’s lory, Toco toucan, and Tucumán amazon. The predominant agent was *P. laurentii* (formerly *C. laurentii*) (25/29; 86.2%), followed by *C. neoformans* (2/29; 6.9%), *F. uniguttulatum* (formerly *C. neoformans* var. *uniguttulatus*) (1/29; 3.4%), and *Solicoccozyma terreus* (formerly *Cryptococcus terreus*) (1/29; 3.4%). However, the contingency table was extremely sparse, with many taxa represented by ≤1–3 submissions and several appearing 100% positive on minimal denominators. Accordingly, these results should be regarded as exploratory and do not provide robust evidence of a stable species-level risk.

#### 3.1.3. Sex

From the 1059 animals analysed in this study, sex information was available for 960 (90.6%). For the remaining 99 animals (9.4%), the requisition form did not specify sex, and these were therefore excluded from the analysis. Among the 960 animals with sex information, 701 were males and 259 were females. No significant association was found between sex and *Cryptococcus* spp. positivity (χ^2^ = 0.12, *df* = 1, *p* = 0.726).

#### 3.1.4. Age

From the 1059 animals analysed in this study, age information was available for 19 of the 48 *Cryptococcus* spp.-positive cases (39.6%), comprising 13 canines and 6 felines. No age data were available for any of the 29 positive avian cases, precluding age-based analysis for this group. Among positive canines, ages ranged from 1 to 19 years, with a median of 4 years (IQR: 2–9.5 years). Positive felines ranged from 3 to 13 years, with a median of 10 years (IQR: 9–13 years). Given the non-normal distribution of age, a Mann–Whitney U test was applied, revealing no statistically significant difference in age between positive and negative animals (U = 1706, z = −1.15, *p* = 0.252, *r* = 0.07).

#### 3.1.5. Geographical Localisation

From the 48 *Cryptococcus* spp.-positive animals, geographical origin (NUTS 2) was available for all cases. Most positives were from the Centre (*n* = 30; 62.5%; 95% CI 48.4–74.8), followed by the North (*n* = 15; 31.3%; 95% CI 20.0–45.3) and Algarve (*n* = 3; 6.3%; 95% CI 2.1–16.8). The distribution of *Cryptococcus* spp. and taxonomically updated related species varied markedly between regions (Table 4). In the North, *C. neoformans* predominated (*n* = 10; 62.5%), whereas in the Centre, *P. laurentii* (formerly *C. laurentii*) was most frequent (*n* = 25; 83.3%). The Algarve cases were exclusively due to *C. neoformans*. No isolates of *Cryptococcus* spp. or related species were recorded in the remaining NUTS 2 regions of Portugal. A chi-square test revealed a statistically significant association between region and *Cryptococcus* species distribution (χ^2^ = 178.66, *df* = 2, *p* < 0.001).

#### 3.1.6. Month and Seasons

Among the 48 *Cryptococcus* spp.-positive animals, the monthly distribution peaked in July (*n* = 9; 18.8%; 95% CI 10.2–31.9), followed by December (*n* = 7; 14.6%; 95% CI 7.2–27.2) and August (*n* = 6; 12.5%; 95% CI 5.9–24.7); June and October each accounted for five cases (10.4%; 95% CI 4.5–22.2). Grouped by astronomical season (Portugal: Winter = 21 December–20 March; Spring = 21 March–20 June; Summer = 21 June–22 September; Autumn = 23 September–20 December), Summer concentrated the largest share of positives (18/48; 37.5%; 95% CI 25.2–51.6) and showed the highest positivity rate relative to the number tested (18/196; 9.2%), followed by Autumn (13/48; 27.1%; 95% CI 16.6–41.0; 13/291; 4.5%), Winter (10/48; 20.8%; 95% CI 11.7–34.3; 10/378; 2.7%), and Spring (7/48; 14.6%; 7/194; 3.6%) (Figure 4). A chi-square test indicated a significant association between month of sampling and case occurrence (χ^2^ = 24.86, *df* = 11, *p* = 0.010).Figure 4Monthly distribution of 48 positive cases of *Cryptococcus* spp. and taxonomically updated related species across diagnostic submissions (2013–2025). Bars depict the monthly positivity rate (per-centage of samples testing positive). The peak occurred in July (14.8%), followed by August (7.8%) and June (6.9%); the lowest rate was observed in January (0.8%).
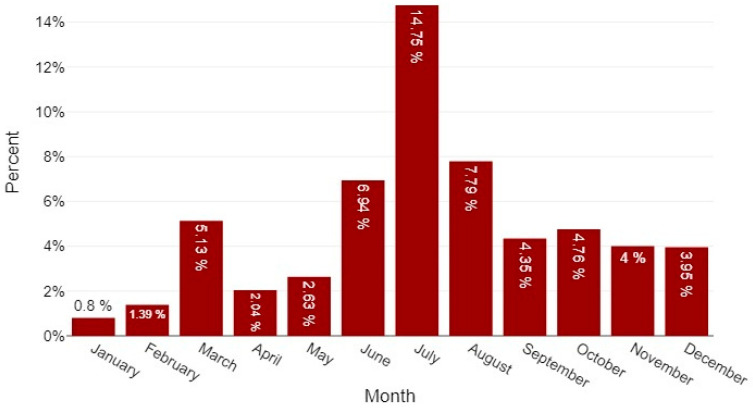


### 3.2. Systematic Review on Cryptococcus spp. in Portugal

A total of 11 studies were included in the final systematic review, selected and reported following the PRISMA 2020 guidelines (Figure 5) [31].

In Europe, cryptococcosis has historically been considered relatively uncommon in domestic animals compared to certain endemic regions such as Australia, the Pacific Northwest of North America, and parts of Africa. However, the increasing recognition of cases in multiple European countries, along with environmental isolations of pathogenic *Cryptococcus* species, suggests that the fungus may be more widely distributed than previously appreciated [32]. Climatic changes, urban expansion, and increased human–animal–environment interaction may facilitate the persistence and spread of these fungi in novel ecological niches [16,33].

Cryptococcosis in Portugal involves both human and animal cases (Table 5), with emerging evidence for the presence of *C. gattii* complex species in the environment. Historically, *C. gattii* was not considered endemic, but a landmark 2007 case involved an immunocompetent Portuguese patient with no travel history diagnosed with *C. gattii* VGII [34]. This was the first autochthonous human case successfully treated with antifungals [34]. Portugal presents a unique context for studying cryptococcosis in domestic animals. The country’s varied climate, from the humid, temperate north to the warmer, drier south, supports diverse vegetation and avian populations, including large urban pigeon colonies and extensive eucalyptus plantations [35]. These environmental features provide potential habitats for both *C. neoformans* and *C. gattii* [16]. In 2015, the first Portuguese feline cryptococcosis case was documented: a 2-year-old stray cat in Vila Real (northern Portugal) developed chronic blepharitis (inflamed, nodular lesions of the eyelid) due to *C. neoformans* [36].

*Cryptococcus* is frequently found in bird excreta and arboreal niches in Portugal [37]. Early surveys in Lisbon (2001) identified urban sources, isolating *Cryptococcus* spp. from pigeon roosting sites [42]. While pigeons rarely develop disease, their dried droppings enrich soil with the fungus, creating an infectious inoculum [43]. A systematic survey in Vila Real (northern Portugal) recovered 28 *C. neoformans* isolates, 23 from pigeon droppings and 5 from decaying eucalyptus leaves [43]. All were *C. neoformans*, though some produced false-positive reactions on CGB agar. Genotyping revealed VNIV (53.6%) as the predominant genotype, followed by VNI (32.1%) and VNIII hybrids (14.3%). These matched certain Portuguese clinical strains, linking environmental and human infections [43]. In 2016, a Mediterranean-wide survey found *C. neoformans* VNI to be the dominant molecular type across the region, with Portugal’s positive isolates originating from tree hollows. No *C. gattii* was recovered from Portuguese sites, reflecting its patchy distribution in Europe. Nevertheless, environmental *C. gattii* was reported in western Spain (Extremadura) in the late 1990s during multiple goat outbreaks in eucalyptus groves, suggesting that similar habitats in southern Portugal, particularly the Alentejo coast, may harbour the fungus. This region shares climate and flora with Spanish endemic zones and was the site of a documented ferret case [35]. Overall, environmental isolates in Portugal are dominated by *C. neoformans* (serotypes A, D, and AD) from both urban and rural sites (Table 6).

## 4. Discussion

This 13-year One Health study combines a PRISMA-guided review of the Portuguese literature with a large veterinary diagnostic series (2013–2025) and provides the first multi-year, laboratory-based picture of *Cryptococcus* spp. in domestic animals across mainland Portugal.

Cryptococcosis is not a notifiable disease in Portugal, either in humans or animals. Consequently, case documentation relies entirely on voluntary reports in the scientific literature. Accordingly, this study reflects passive surveillance of clinically affected animals rather than systematic screening of the general companion animal population, as all specimens were submitted in response to compatible clinical signs documented on requisition forms. This approach increases the pre-test probability of true infection but precludes reliable inference of prevalence in the wider Portuguese animal population.

The scarcity of data is evident, with only seven scientific reports on cryptococcal infections in humans and animals in Portugal published over the past 25 years. Within this context, our retrospective study addresses the evidence gap by extending and integrating previous Portuguese case reports and environmental isolations, yielding insights that are consonant with the known ecology and epidemiology of the *Cryptococcus* species complexes in Europe and elsewhere [1,2,5,6,14,16,43].

### 4.1. Cryptococcus Species Distribution and Host Compartments

The predominance of *C. neoformans* among canine and feline cases in our series (73.7% of carnivore positives) is consistent with its well-recognised association with avian-derived, urban environments (particularly pigeon droppings) and its capacity to cause disease in immunocompromised hosts and companion animals [14,16,23]. By contrast, the striking representation of *P. laurentii* (formerly *C. laurentii*) among avian culture-positives (86.2%) may reflect two non-exclusive explanations: (a) *P. laurentii* (and related non-*C. neoformans* species) may be more frequent colonisers or opportunistic pathogens in captive/commensal birds within our submission population, and (b) sampling bias—many avian submissions derived from pet or aviary birds kept under husbandry conditions that favour exposure to environmental yeasts. Previous Portuguese environmental surveys identified *C. neoformans* in pigeon droppings and tree debris [43,46], but veterinary avian disease may result from colonisation or secondary infection by non-major pathogenic species, a pattern reported in veterinary mycology literature [5,21].

Interpretation also varies by anatomical site, as isolation from normally sterile compartments (e.g., CSF, tissue aspirates, sterilely collected urine) provides strong evidence of invasive cryptococcosis in affected carnivores, whereas growth from non-sterile mucosal sites, particularly in birds, may represent colonisation or transient carriage [5,14,21,23,29].

Cryptococcosis affects multiple host species, and our findings confirm the multispecies nature of cryptococcosis, with culture-confirmed infections in dogs, cats, and several avian species. Clear interspecies differences in aetiological distribution were evident: *C. neoformans* predominated in carnivores, whereas *P. laurentii* (formerly *C. laurentii*) was most frequent in birds. These host–pathogen patterns are in line with previous Portuguese environmental data and with broader veterinary mycology literature [5,21].

Susceptibility seems to vary by host species and individual health status. In human medicine, HIV-associated immunosuppression is a well-established risk factor [36], and analogous feline retroviral infections (feline immunodeficiency virus [FIV] and feline leukaemia virus [FeLV]) may plausibly modulate increased susceptibility in cats. In the present dataset, however, FIV/FeLV status was rarely available and could not be systematically assessed. Prospective studies incorporating host immune parameters are warranted to clarify species and health state-specific vulnerabilities and to refine veterinary risk stratification.

### 4.2. Ecological and Geographic Patterns

The regional clustering—a central region predominated by *P. laurentii* (formerly *C. laurentii*) and a northern region where *C. neoformans* prevailed—suggests spatial heterogeneity in environmental reservoirs or referral/submission patterns to the diagnostic laboratory. Portugal’s varied climates and land use (urban pigeon aggregations, eucalyptus stands, and agricultural mosaics) provide plausible micro-habitats favourable to different *Cryptococcus* taxa [35,43]. The systematic review portion of this work corroborates prior autochthonous *C. gattii* complex reports in Portugal and a history of environmental *C. neoformans* in pigeon droppings and tree hollows [34,35,43]; our veterinary series, therefore, likely reflects a mixture of true ecological differences plus referral bias (veterinary practices in some regions may be more likely to submit mycological samples).

### 4.3. Seasonality and Transmission Implications

The observed summer peak (highest positivity and proportion of positives in samples collected between June and September) is biologically plausible: warmer, drier months facilitate aerosolisation of desiccated yeast cells and basidiospores from environmental reservoirs (e.g., dried droppings, decaying tree matter), increasing inhalational exposure risk for animals and humans alike [14,15]. A strong monthly/seasonal signal supports the value of targeted environmental surveillance and public-health messaging during higher-risk periods (e.g., advising owners of susceptible animals or immunocompromised people about exposure reduction in summer months).

Nevertheless, as this is a convenience series of diagnostic submissions, seasonal and regional signals may also reflect referral and sampling patterns. These ecological interpretations should therefore be regarded as hypothesis-generating rather than population-representative.

### 4.4. Diagnostic Approach and Laboratory Considerations

Use of routine culture combined with MALDI-TOF MS allowed species-level identification in most isolates and makes this dataset operationally useful. MALDI-TOF MS proved valuable for high-throughput identification in a diagnostic setting, but its performance is contingent on the completeness of reference libraries and the inclusion of non-*C. neoformans/C. gattii* taxa (e.g., *P. laurentii* (formerly *C. laurentii*), and *N. humicola* (formerly *C. humicola*)) in the database. The identification of several non-*neoformans* species highlights the need for expanded reference spectra and, where clinically or epidemiologically relevant, confirmation/typing by molecular or serological methods. Importantly, species-level identification by MALDI-TOF MS alone does not provide molecular type information (e.g., VNI, VGII), which has proven epidemiological and clinical relevance (especially for *C. gattii* complex infections) [22,34].

Cryptococcal antigen (CrAg) assays, although widely applied in human medicine, are also used in veterinary diagnostics to support rapid presumptive diagnosis. However, CrAg positivity reflects infection with the *C. neoformans/C. gattii* complex but does not provide species or genotype resolution, nor does it reliably distinguish between active infection and subclinical carriage. Indeed, reports from outbreak settings indicate CrAg positivity in clinically normal animals (e.g., 7.1% of cats and 0.8% of dogs) during a *C. gattii* outbreak [47], highlighting that antigen detection is not synonymous with overt disease. Routine CrAg testing was not systematically available in this retrospective dataset, which represents a recognised limitation and an area for methodological enhancement in future studies.

Quantitative culture methods add interpretability, particularly for sterile fluids such as CSF and blood, where higher fungal burdens (e.g., CFU/mL) correlate with adverse outcomes and can be monitored longitudinally [48,49,50]. Although culture remains the diagnostic reference, and any growth from a normally sterile site is clinically significant, colony counts were not routinely recorded and could not be analysed.

Finally, interpretation must be contextualised by anatomical site and clinical presentation. Isolation from normally sterile compartments (e.g., CSF, tissue aspirates, and sterilely collected urine) provides strong evidence of invasive cryptococcosis, whereas growth from non-sterile mucosal sites, particularly in birds, may represent colonisation or transient carriage. Within a One Health framework, both colonised and infected animals serve as ecological sentinels of environmental *Cryptococcus* exposure, but inference of prevalence or disease burden must remain cautious [51,52].

### 4.5. One Health, Planetary Health, and Public Health Relevance

Although direct animal-to-human transmission of *Cryptococcus* is considered rare, infected animals serve as important sentinels of local environmental exposure and can contribute to the identification of environmental hotspots of concern for human health, particularly among immunocompromised individuals [6,22,52].

Background colonisation in shelter/pound populations can be substantial, with consolidated evidence reporting upper-bound estimates of approximately 25% in dogs and 20% in cats, and most carriers remaining unaffected [18,47,53,54,55].

Detection of *C. gattii* complex infections in Portuguese animals and humans confirms the occurrence of autochthonous exposure and underscores the need for integration between veterinary surveillance and public health monitoring to better characterise risks and sources [34,41]. This study’s combination of laboratory-based veterinary investigations and a systematic literature review illustrates how routine diagnostic data may complement environmental and clinical research within a One Health and Planetary Health framework.

### 4.6. Limitations of This Study

The retrospective laboratory design and reliance on passive surveillance data impose several important constraints. First, clinical submissions are subject to referral and sampling bias: the denominator (animals tested) does not uniformly represent the at-risk population across regions, species, or husbandry systems. Second, metadata were incomplete for many records (e.g., age information was missing for most positives, and antifungal treatment/outcome data were sparse), limiting both risk-factor analysis and clinical outcome assessment. Third, although MALDI-TOF MS enabled rapid species-level identification, molecular genotyping (AFLP, MLST) and antifungal susceptibility testing were not performed for most isolates. Such data are crucial to link clinical and environmental strains, detect introductions or expansions of particular molecular types (e.g., *C. gattii* VGII), and monitor potential shifts in antifungal susceptibility. Fourth, environmental sampling was not conducted concurrently or systematically alongside the animal case series, restricting direct source attribution. Finally, the relatively small absolute number of positives (*n* = 48) constrains statistical power and the generalisation of breed- or species-level associations; several apparent breed associations were driven by small denominators and should therefore be interpreted with caution.

In addition, the absence of systematic antigen testing (serum/CSF) limited the ability to reliably differentiate colonisation from active infection in non-sterile site positives. Future studies should incorporate routine CrAg testing together with molecular typing to strengthen diagnostic resolution. Likewise, colony counts were not recorded in routine culture reports, although any growth from a normally sterile site is clinically significant; the absence of quantitative culture precluded burden-based analyses and longitudinal monitoring.

Despite these limitations, this work includes the long period (13 years), the combined One Health literature review, the species-level identifications from a high-volume veterinary diagnostic laboratory, and the attempt to quantify seasonal and regional trends. Despite limitations inherent to passive surveillance, the consistent detection of *Cryptococcus* spp. over more than a decade indicates persistent environmental reservoirs and sporadic clinical impact in domestic animals.

### 4.7. Recommendations and Future Directions

To strengthen surveillance and risk mitigation in Portugal, we recommend the following:Routine molecular typing (MLST/AFLP or whole-genome sequencing where feasible) of clinical and environmental *Cryptococcus* isolates to map molecular types and detect emergent strains.Systematic, seasonally stratified environmental sampling (pigeon roosts, eucalyptus and other tree hollows, soil, and compost) in regions with veterinary or human cases to identify reservoirs and temporal dynamics.Integration of veterinary diagnostic data with human public-health laboratories (shared databases and joint investigations) under a One Health umbrella.Incorporation of antifungal susceptibility testing for clinical isolates to inform veterinary therapy and to monitor resistance trends.Targeted education for veterinarians and pet owners, particularly during summer months, on cryptococcosis recognition and strategies to reduce environmental exposure.

From a laboratory perspective, expanding and validating MALDI-TOF MS reference libraries for non-*neoformans* species and establishing reflex molecular typing for unusual or epidemiologically important isolates will increase the public-health utility of routine diagnostics.

Future protocols should also prioritise semi-quantitative or quantitative cultures with harmonised reporting standards to improve prognostic assessments and strengthen comparative epidemiological analyses.

## 5. Conclusions

This One Health systematic review and 13-year veterinary laboratory series document the endemic presence of *Cryptococcus* spp. in Portugal, with notable host compartmentalisation: *C. neoformans* predominates in feline and canine clinical cases, while *Papiliotrema laurentii* (formerly *C. laurentii*) was most frequently isolated from avian submissions. Cases were concentrated in the Centre and North regions and were more frequent in the summer months. As cryptococcosis is not notifiable in Portugal, surveillance remains passive and inherently incomplete. These findings support the need for routine molecular typing, antifungal susceptibility testing, and integrated human–animal–environment surveillance to clarify environmental sources, detect and characterise *C. gattii* complex occurrences, and inform prevention and clinical management. Prospective, systematic environmental sampling and coordinated One Health data sharing will be the essential next steps to translate these descriptive insights into effective veterinary and public-health interventions.

## Figures and Tables

**Figure 1 jof-11-00672-f001:**
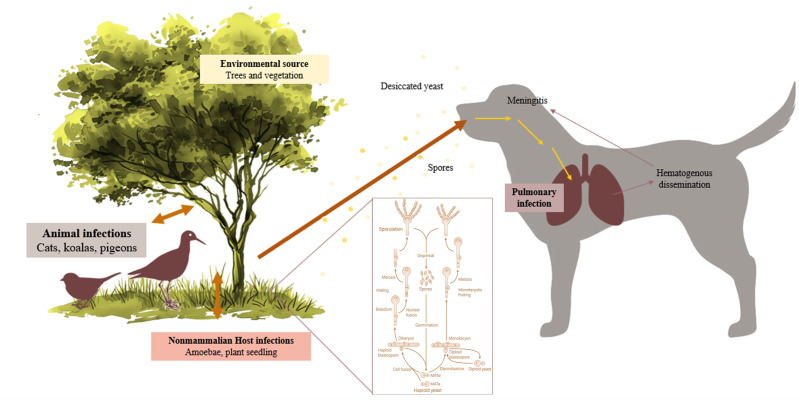
Life cycle of *Cryptococcus* spp.

**Figure 2 jof-11-00672-f002:**
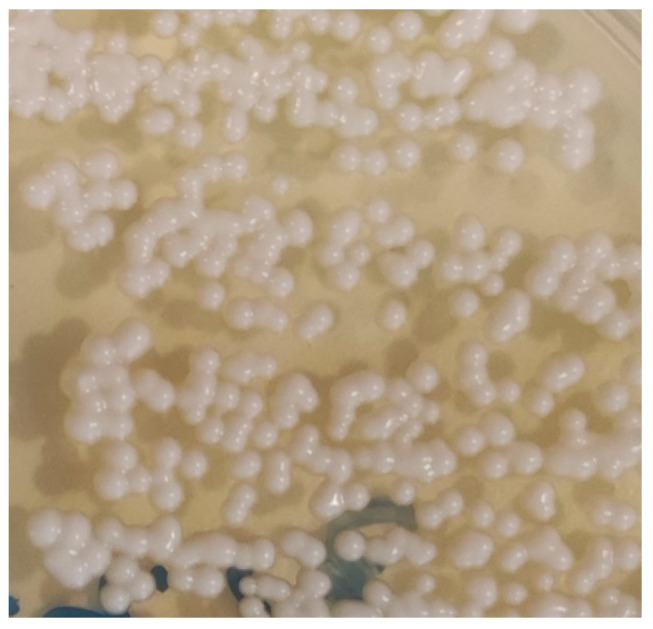
*Cryptococcus* spp. colonies onto Sabouraud Dextrose Selective Agar, Emmons formulation, containing chloramphenicol and gentamicin (Thermo Scientific™, R01772, Waltham, MA, USA), incubated at 30 °C.

**Figure 3 jof-11-00672-f003:**
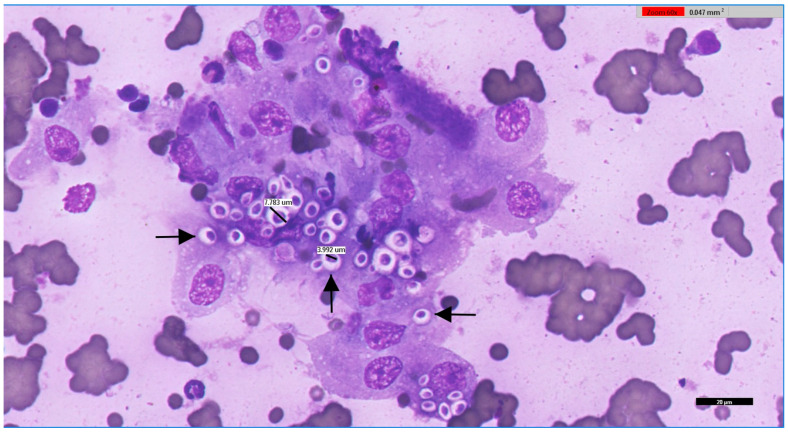
Renal cryptococcoma in a European Shorthair cat: fine-needle aspirate cytology (May–Grünwald–Giemsa). Fine-needle aspirate from a renal mass shows numerous macrophages containing *Cryptococcus neoformans* (arrows). Yeast cells are round to oval, occasionally elongated, with variable size both with and without the capsule. The thick polysaccharide capsule appears colourless to pale pink, while the organisms stain pink–purple and occasionally exhibit narrow-based budding (middle arrow). The background contains moderate numbers of erythrocytes and mixed inflammatory cells, including neutrophils, lymphocytes, and occasional eosinophils. A digital whole-slide scan was obtained using Philips IntelliSite Pathology Solution 3.2 (Philips Medical Systems Netherlands B.V., 2017). Scale bar = 20 μm.

**Figure 5 jof-11-00672-f005:**
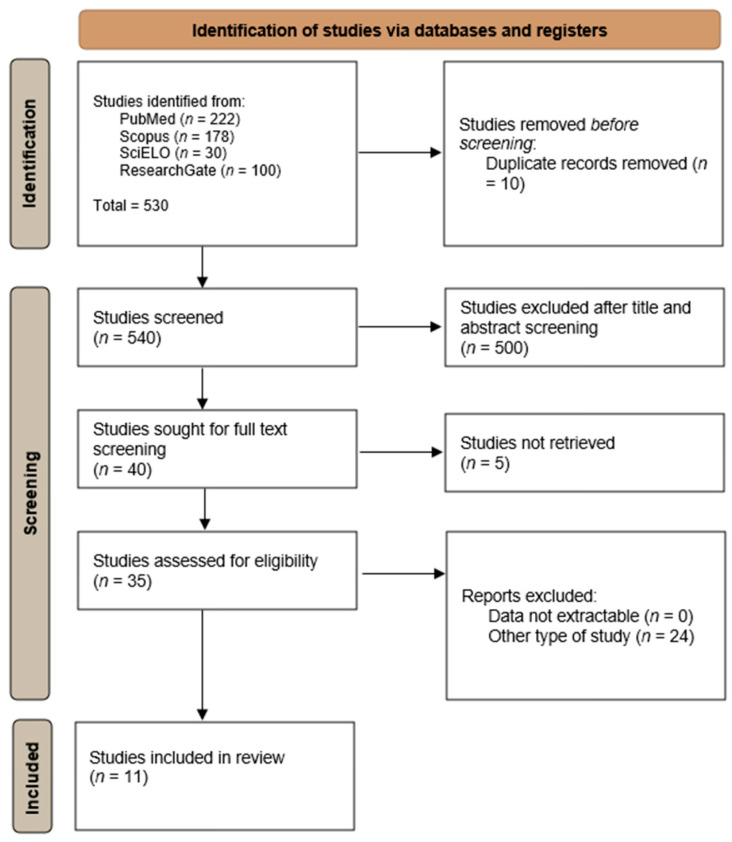
Flow diagram of the identification and extraction process of articles included in this study (adapted from Prisma guidelines 2020) [31].

**Table 1 jof-11-00672-t001:** Distribution of *Cryptococcus* spp. and taxonomically updated related species-positive cases by animal species in Portugal, 2013–2025.

Distribution of *Cryptococcus* spp.-Positive Cases by Animal Species							
	Negative			Positive			Total
Species	*n*	% Within Species	95% CI (%)	*n*	% Within Species	95% CI (%)	*n*
Avian	691	96.0%	94.3–97.2	29	4.0%	2.8–5.8	720
Canine	177	93.2%	88.7–96.0	13	6.8%	4.0–11.4	190
Feline	107	94.7%	88.9–97.5	6	5.3%	2.5–11.1	113
Equine	9	100%	0.7–100	0	0%	-	9
Lagomorphs	7	100%	64.6–100	0	0%	-	7
Rodents	1	100%	20.7–100	0	0%	-	1
Other mammals	8	100%	67.6–100	0	0%	-	8
Non-mammalian non-avian vertebrates	11	100%	67.6–100	0	0%	-	11
Total	1011	95.6%	94.0–96.6	48	4.5%	3.4–6.0	1059

CI, confidence interval. Other mammals include non-rodents and non-lagomorphs. Non-mammalian, non-avian vertebrates include chelonians, other reptiles, and fish.

**Table 2 jof-11-00672-t002:** Distribution of *Cryptococcus* spp. and taxonomically updated related species among positive canine and feline cases by breed in Portugal, 2013–2025.

		*Papiliotrema * *laurentii * (Formerly *C. laurentii *)		*Filobasidium uniguttulatum *(Formerly *C. neoformans* var. *uniguttulatus*)		*Cryptococcus * *neoformans*		*Naganishia humicola *(Formerly *C. humicola*)		Total
	Breeds	*n*	% Within Breed	*n*	% Within Breed	*n*	% Within Breed	*n*	% Within Breed	*n*
**Canine**	Boston Terrier	0	0%	0	0%	2	100%	0	0%	2
	Dobermann	0	0%	0	0%	0	0%	1	100%	1
	French Bulldog	1	100%	0	0%	0	0%	0	0%	1
	German Shepherd	0	0%	0	0%	1	100%	0	0%	1
	Saint Bernard	0	0%	0	0%	1	100%	0	0%	1
	Siberian Husky	1	100%	0	0%	0	0%	0	0%	1
	Yorkshire Terrier	0	0%	0	0%	1	100%	0	0%	1
	Mixed Breed	1	14.3%	1	14.3%	5	71.4%	0	0%	7
**Feline**	European Shorthair	0	0%	0	0%	4	100%	0	0%	4
	Total	3	15.8%	1	5.3%	14	73.7%	1	5.3%	19

Percentages are calculated within each breed. Totals represent the proportion of each *Cryptococcus* spp. and taxonomically updated related species among all positive cases (*n* = 19).

**Table 3 jof-11-00672-t003:** Distribution of *Cryptococcus* spp. and taxonomically updated related species among positive avian cases by host species in Portugal, 2013–2025.

	*Papiliotrema * *laurentii * (Formerly *C. laurentii *)		*Filobasidium uniguttulatum *(Formerly *C. neoformans* var. *uniguttulatus*)		*Cryptococcus * *neoformans*		*Solicoccozyma terreus *(Formerly *C. terreus)*		Total
Avian Species	*n*	% Within Species	*n*	% Within Species	*n*	% Within Species	*n*	% Within Species	*n*
African grey parrot (*Psittacus erithacus*)	1	50%	0	0%	0	0%	1	50%	2
Atlantic canary (*Serinus canaria*)	5	83.3%	1	16.7%	0	0%	0	0%	6
Blue-and-yellow macaw (*Ara ararauna*)	1	100%	0	0%	0	0%	0	0%	1
Budgerigar (*Melopsittacus undulatus*)	1	100%	0	0%	0	0%	0	0%	1
Chestnut-billed toucanet (*Aulacorhynchus castaneotis*)	1	100%	0	0%	0	0%	0	0%	1
Cockatiel (*Nymphicus hollandicus*)	0	0%	0	0%	1	100%	0	0%	1
Eclectus parrot (*Eclectus roratus*)	1	100%	0	0%	0	0%	0	0%	1
Emerald toucanet (*Aulacorhynchus prasinus*)	3	100%	0	0%	0	0%	0	0%	3
European goldfinch (*Carduelis carduelis*)	1	100%	0	0%	0	0%	0	0%	1
Lovebird (*Agapornis* spp.)	3	100%	0	0%	0	0%	0	0%	3
Parakeet (*Cyanoramphus* spp.)	1	100%	0	0%	0	0%	0	0%	1
Red lory (*Lorius lory erythrothorax*)	1	100%	0	0%	0	0%	0	0%	1
Red-crowned parakeet (*Cyanoramphus novaezelandiae*)	1	100%	0	0%	0	0%	0	0%	1
Ring neck (*Psittacula krameri*)	1	100%	0	0%	0	0%	0	0%	1
Rock pigeon (*Columba livia*)	1	100%	0	0%	0	0%	0	0%	1
Society finch (*Lonchura striata*)	1	100%	0	0%	0	0%	0	0%	1
Stella’s lory (*Lorius stella*)	1	100%	0	0%	0	0%	0	0%	1
Toco toucan (*Ramphastos toco*)	1	100%	0	0%	0	0%	0	0%	1
Tucumán amazon (*Amazona tucumana*)	0	0%	0	0%	1	100%	0	0%	1
Total	25	86.2%	1	3.5%	2	6.9%	1	3.5%	29

Percentages are calculated within each avian host species. Totals represent the proportion of each *Cryptococcus* spp. and taxonomically updated related species among all positive avian cases (*n* = 29).

**Table 4 jof-11-00672-t004:** Regional (NUTS 2) distribution of *Cryptococcus* spp. and taxonomically updated related species among positive cases in Portugal, 2013–2025.

	*Papiliotrema laurentii *(Formerly *C. laurentii *)		*Filobasidium uniguttulatum *(Formerly *C. neoformans* var. *uniguttulatus *)		*Cryptococcus * *neoformans*		*Naganishia humicola *(Formerly *C. humicola*)		*Solicoccozyma terreus *(Formerly *C. terreus *)		Total
Region(NUTS2)	*n*	% Within Region	*n*	% Within Region	*n*	% Within Region	*n*	% Within Region	*n*	% Within Region	*n*
North	3	10.7%	1	50%	10	62.5%	1	100%	0	0%	15
Centre	25	89.3%	1	50%	3	18.8%	0	0%	1	100%	30
Algarve	0	0%	0	0%	3	18.8%	0	0%	0	0%	3
Total	28	100%	2	100%	16	100%	1	100%	1	100%	48

**Table 5 jof-11-00672-t005:** Reported animal and human cryptococcosis cases in Portugal (2000–2025).

Species	Year	Location	*Cryptococcus* spp. (Strain)	Clinical Manifestation	Outcome	Ref.
HIV/AIDS human patients	2000s	Lisbon	*C. neoformans* (VNI, VNII, VNIII, VNIV)	Meningoencephalitis	Some recovery, some death	[37]
Middle-aged man, male	2007	Lisbon	*C. gattii* (VGII)	Pulmonary and systemic	Recovery	[34]
Domestic ferret (*Mustela putorius furo*)	2014	V. N. Milfontes	*C. gattii* VGIII (AFLP6)	Pulmonary cryptococcosis (nodular masses in the lung)	Euthanised	[38]
Goat (Bravia breed)	2014	Lisbon	*C. neoformans*	Cryptococcal meningitis (post mortem diagnosis)	Death	[39]
Domestic cat (*Felis catus*)	2015	Vila Real	*C. neoformans*	Ocular cryptococcosis (blepharitis)	Recovery	[36]
Adult man, male	2019	Azores	*C. deuterogattii* (VGII)	Lung cryptococcomas, CNS lesions	Recovery	[40]
African grey parrot (*Psittacus erithacus*)	2021	Almada	*C. bacillisporus* VGIII (AFLP5)	Rhinothecal cryptococcoma	Recovery	[41]

AFLP, amplified fragment length polymorphism genotype; VG, *C. gattii* molecular type.

**Table 6 jof-11-00672-t006:** Environmental isolation studies of *Cryptococcus* in Portugal (2000–2025).

Region	Year	Sample Source	*Cryptococcus* spp. (Strain)	Ref.
Lisbon	2001	Pigeons’ roost droppings	*C. neoformans*	[44]
Mediterranean region	2012–2015	Tree hollows, trunks, bark, and soil	*C. neoformans*	[35]
Vila Real	2014	Pigeon droppings; eucalyptus tree detritus	*C. neoformans* (VNIV, VNI, VNIII)	[43]
Various sites in Portugal	2016	Tree hollows	*C. neoformans* (VNI)	[45]

## Data Availability

The data presented in this study are available upon request from the corresponding authors.

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
