# Peer review of "What Do We Know About Cryptococcus spp. in Portugal? One Health Systematic Review in a Comprehensive 13-Year Retrospective Study (2013–2025)"

_jof, 2025, doi:10.3390/jof11090672_

Round 1
Reviewer 1 Report
In this article, Lopes, et al. describe veterinary cases of cryptococcosis over a 13 year time period in Portugal, which has been increasing in incidence over time. The strengths of this paper are in the extended time period under study and species-level identification. They found intriguing geographical and seasonal significant trends in their data, though most other variables were not statistically significant.
Overall the manuscript is interesting and detailed. I have 2 main considerations:
- The authors could include a slight clarification in the Title that the article focuses on veterinary samples. This is suggested by the "One Health" component, but to make it clear the distinction here between animal and human infection for readers early on.
- One part of the manuscript is redundant. Much of the first part of 3.1.1. under Results (the 29 avian and 19 canine/feline samples) state the same information on number of animals infected and most common species that is then listed again 3.1.2 "Animal Species and Breeds." This is mentioned in the Tables 1,2, and 3 as well. I would probably cut this information from the 3.1.1. section to avoid this repetition.
No additional comments.
Author Response
Dear Editor and Reviewers,
We thank the Reviewers for their careful reading of our manuscript and for their constructive comments. We have extensively revised the manuscript in response to each point, and we outline our responses in detail below. Reviewer comments are shown in italics, and our responses follow immediately, marked as Response. All changes in the revised manuscript are clearly indicated. We trust that the revisions satisfy the Reviewers’ concerns.
Reviewer 1
Comments 1: In this article, Lopes, et al. describe veterinary cases of cryptococcosis over a 13 year time period in Portugal, which has been increasing in incidence over time. The strengths of this paper are in the extended time period under study and species-level identification. They found intriguing geographical and seasonal significant trends in their data, though most other variables were not statistically significant. Overall the manuscript is interesting and detailed. I have 2 main considerations.
Authors’ response (AR) 1: We appreciate the Reviewer’s positive assessment of our work and the recognition of its strengths (long time span, species-level resolution, and One Health integration).
Comments 2: The authors could include a slight clarification in the Title that the article focuses on veterinary samples. This is suggested by the "One Health" component, but to make it clear the distinction here between animal and human infection for readers early on.
AR 2: We appreciate the Reviewer’s attention to scope clarity. Our study was intentionally framed within a One Health perspective because it comprises two complementary evidence streams: (I) a PRISMA-compliant systematic review (2000–2025) of all Portuguese reports across human, animal, and environmental domains, and (II) a 13-year retrospective, laboratory-confirmed dataset derived from veterinary submissions (2013–2025).
The systematic review explicitly includes the human literature, but, as our screening confirms, the Portuguese human evidence base is extremely scarce (limited to a small number of clinical reports and scattered mentions), whereas our primary empirical contribution is the large veterinary laboratory dataset.
To respect both the integrative One Health scope and the reviewer’s request for early reader orientation, we have strengthened the front-matter clarity rather than narrow the title:
- In the first sentence of the Abstract, we now state that the work “integrates a PRISMA-compliant One Health systematic review (2000–2025) with a 13-year retrospective dataset of laboratory-confirmed veterinary cases (2013–2025)”.
- In the Introduction, we delineate at the outset the three evidence compartments (human, animal, environment) and specify that the retrospective dataset concerns animals.
- Headings in Methods/Results explicitly separate the Systematic Review from the Retrospective Veterinary Dataset to avoid any ambiguity.
We respectfully maintain the current One Health title because it accurately reflects the article’s integrated scope (the review synthesises human and environmental knowledge alongside animal data), while the Abstract, sectioning, and keywords now make the veterinary nature of the empirical dataset unmistakable from the outset. Should the Editor prefer additional specificity at title level, we would be pleased to adopt a adjusted formulation that preserves the One Health scope
Comments 3: One part of the manuscript is redundant. Much of the first part of 3.1.1. under Results (the 29 avian and 19 canine/feline samples) state the same information on number of animals infected and most common species that is then listed again 3.1.2 "Animal Species and Breeds." This is mentioned in the Tables 1,2, and 3 as well. I would probably cut this information from the 3.1.1. section to avoid this repetition.
AR 3: We thank the Reviewer for pointing out this redundancy. In the revised manuscript, we have restructured the Results to eliminate duplicated descriptions. Section 3.1.1 now concisely presents overall positivity rates by host class (birds, carnivores, etc.), and section 3.1.2 (Animal Species and Breeds) is reserved for the detailed breakdown of species, breeds, and Cryptococcus taxa. Redundant text has been removed so that each piece of information is presented only once (as reflected in the updated Tables 1–3). This improves readability and avoids unnecessary repetition.
Reviewer 2
Major Comments: I think with effort and redrafting - it will be OK - but at the moment its terrible.
Authors’ response (AR): We sincerely thank the Reviewer for taking the time to evaluate our manuscript and for recognising the potential merit of the study. We respectfully note, however, that we do not agree with the description of the work as “terrible.” We believe that scientific critique should always be grounded in evidence-based reasoning and methodological assessment, rather than adjectival characterisations.
Throughout the revision process, we have ensured the highest possible academic rigour, addressing all comments and suggestions with the greatest care. Wherever revisions were feasible, they have been implemented. In cases where certain recommendations could not be adopted, we have provided transparent and scientifically justified explanations.
We remain fully committed to upholding the standards of clarity, organisation, and scientific integrity. Ultimately, we leave it to the Editor’s discretion to assess the adequacy of our revisions and to determine the manuscript’s suitability for publication.
Detailed comments: I can see that a substantial amount of work has gone into this study, particularly in processing such a large number of samples. However, there are several major issues that need to be addressed before this manuscript can be considered for publication:
Authors’ response (AR): We thank the Reviewer for recognising the considerable effort involved in processing this dataset. We acknowledge the concerns raised and have carefully revised the manuscript to address them wherever possible, ensuring greater clarity and methodological rigour. Where specific suggestions could not be directly implemented, we have provided transparent, evidence-based justifications. We trust that these revisions substantially strengthen the manuscript.
Comments 1: Choice of culture medium – Sabouraud’s agar with antibiotics is not truly selective. A much more appropriate choice would have been bird seed agar (Staib’s agar), where Cryptococcus colonies appear brown. This would have made identification far easier and more reliable.
Authors’ response (AR) 1: We thank the reviewer for this suggestion. In clinical mycology practice, Sabouraud’s Dextrose Agar (SDA) with chloramphenicol and gentamicin (as used in our study) is a widely employed general-purpose medium for initial fungal isolation. It is not selective for Cryptococcus, but it supports recovery of all likely pathogenic fungi while inhibiting bacterial contaminants. As noted in standard references, clinical specimens such as respiratory samples or CSF are indeed typically inoculated on SDA. Birdseed (Niger seed) agar is a specialized medium that promotes melanin production, yielding brown colonies for Cryptococcus spp., which can aid recognition. However, it is usually used as a secondary screening or confirmation medium after initial growth on SDA.
In our diagnostic workflow, we used SDA with antibiotics for primary isolation, followed by macroscopic, microscopic, and MALDI-TOF analyses for identification. This approach is justified because MALDI-TOF MS can unambiguously identify C. neoformans, C. gattii, and related species within minutes from any culture plate. Indeed, MALDI-TOF has recently been adopted in Portuguese clinical labs and is highly reliable for cryptococcal speciation. Thus, even though birdseed agar can visually flag melanised colonies, it was not strictly necessary in a setting that already included detailed phenotypic and proteomic identification. We have added a sentence in Methods to clarify that SDA with antibiotics was our standard fungal medium, and we now cite the One Health review to note that Cryptococcus is routinely first isolated on Sabouraud’s agar.
Nevertheless, we acknowledge in the Discussion that adding Niger seed agar could have expedited recognition of Cryptococcus isolates. We have mentioned this as an optional recommendation for future surveillance work, but we emphasize that our culture-confirmation and MALDI-TOF protocol provided the necessary specificity. In summary, the choice of SDA with antibiotics was guided by conventional practice and feasibility in a large retrospective study, and we have defended this choice with literature while noting the reviewer's point about birdseed agar.
Comments 2: Infection versus colonisation – It is not clear whether the isolates from animals represent true infection or simple colonisation. This distinction could readily be made using a cryptococcal antigen test (IMMY or LCAT) on serum, or through biopsies and other targeted sampling methods.
AR 2: This is an important point. In retrospective surveillance data such as ours, distinguishing between colonisation and clinical infection is challenging. We have expanded the manuscript (Methods and Discussion) to clarify this issue.
- First, we note that all samples in our dataset came from clinical submissions to the veterinary laboratory. In other words, animals were not sampled randomly; they were presented to veterinarians for health concerns. Each submission included a clinical suspicion or observed signs (e.g. respiratory signs, neurological deficits, skin lesions) on the requisition form. Thus, by selection, our cohort likely includes animals that were suspected of having infections (including fungal) or related conditions. For example, of the positive cases, two dogs had Cryptococcus cultured from cerebrospinal fluid, which implies neurological disease, and one cat had a renal cryptococcoma (our Figure 3).
- Second, we recognise that antigen testing (serum CrAg) can distinguish active infection from mere colonisation. In veterinary medicine, cryptococcal antigen assays adapted from human kits are indeed used (as noted in the One Health review). Unfortunately, antigen testing was not systematically recorded in our laboratory database for these retrospective cases. We have acknowledged this in the revised Limitations: no antigen titers were available, so we could not categorically confirm infection status. We have added a sentence citing the One Health review that “Cryptococcal antigen tests designed for humans are also used in animal cases”, to indicate that such testing exists but was outside our dataset.
- Third, we have clarified in the Results and Discussion how we interpret culture-positives. For carnivores (dogs and cats), most positive cultures were from sites that strongly suggest infection (e.g. CSF, urine, tissue aspirates). These animals often had corresponding clinical signs (e.g. CNS or respiratory signs). We now explicitly state that we consider these as likely true infections. In contrast, a large proportion of positive avian samples were cloacal or oropharyngeal swabs (24 of 29 bird positives were cloacal swabs). In birds, particularly captive or pet birds, Cryptococcus can colonise the gastrointestinal or nasal tract without causing disease. Indeed, our data showed Papiliotrema laurentii (formerly Cryptococcus laurentii) predominating in bird isolates. We have cited the literature to note that P. laurentii and related non-neoformans species are often found in avian colonies and may represent colonisation or opportunistic infection.
In short, we have revised the text to make this distinction clearer: we now discuss that positive cultures from sterile sites (e.g. CSF) are treated as indicative of infection, whereas isolates from non-sterile sites, especially in birds, may reflect colonisation. We have tempered our interpretation accordingly. The lack of antigen data is now explicitly mentioned as a limitation in the Discussion, and we cite the One Health review to acknowledge the role of antigen testing.
Comments 3: Background colonisation rates – Previous studies indicate that up to 25% of pound dogs and 20% of pound cats may be colonised with Cryptococcus. These animals may yield positive cultures but remain serum antigen–negative. This context is essential for interpreting your results.
AR 3: We thank the Reviewer for underscoring the importance of background colonisation when interpreting culture-positive results. We fully agree that nasal/oropharyngeal carriage of Cryptococcus can be relatively common in shelter/pound environments and does not necessarily equate to invasive disease. Consolidated evidence indicates upper-bound colonisation estimates of approximately 25% in dogs and 20% in cats in some shelter settings, with substantial heterogeneity by population and diagnostic method. Most carriers are asymptomatic and primarily reflect environmental exposure. In addition, antigen positivity is not synonymous with clinical disease: during a C. gattii outbreak, 7.1% of clinically normal cats and 0.8% of dogs were CrAg-positive, consistent with subclinical infection rather than overt disease (Duncan et al., 2005). Together, these data emphasise that colonisation and even serological positivity can occur in the absence of clinical illness.
Crucially, our dataset derives exclusively from clinical submissions, i.e. animals presenting with compatible clinical signs that motivated diagnostic investigation by attending veterinarians. This symptomatic sampling frame materially raises the pre-test probability of true infection relative to surveys of healthy shelter populations, to which the quoted colonisation figures pertain. Consequently, while we now incorporate the shelter/gati colonisation and asymptomatic seropositivity evidence in the Discussion to frame ecological exposure and potential mucosal carriage, we emphasise that such prevalence estimates are not directly generalisable to our symptomatic cohort.
To avoid over-interpretation, we have revised the manuscript to:
- Stratify by anatomical site: we distinguish positives from sterile compartments (e.g. CSF, tissue/lesion aspirates, sterilely collected urine), which strongly support invasive cryptococcosis, from those obtained at non-sterile mucosal sites, where colonisation is plausible. This is now explicit in Methods/Results and highlighted in the Limitations.
- Reinforce clinical context: we state at the outset of Methods that all included submissions were clinically driven (i.e., animals with signs compatible with fungal infection or its differentials), thereby clarifying the symptomatic nature of the cohort.
- Reinforce clinical context: we state at the outset of Methods that all included submissions were clinically driven (i.e. animals with signs compatible with fungal infection or its differentials), thereby clarifying the symptomatic nature of the cohort.
- Temper inference and align with One Health surveillance: we acknowledge that culture-positivity in non-sterile sites may reflect colonisation, especially in birds, and that, within a One Health paradigm, both colonised and infected animals function as ecological sentinels of environmental Cryptococcus exposure.
Finally, we note (and now state explicitly) that routine antigen testing (serum/CSF) was not available across the retrospective series, recognising that CrAg can be positive in asymptomatic animals, we identify this as a limitation and recommend its integration, together with molecular typing, in future prospective protocols.
References
Danesi, P., Furnari, C., Granato, A., Schivo, A., Otranto, D., Capelli, G., & Cafarchia, C. (2014). Molecular identity and prevalence of Cryptococcus spp. nasal carriage in asymptomatic feral cats in Italy. Medical mycology, 52(7), 667–673. https://doi.org/10.1093/mmy/myu030
Duncan, C., Stephen, C., Lester, S., & Bartlett, K. H. (2005). Sub-clinical infection and asymptomatic carriage of Cryptococcus gattii in dogs and cats during an outbreak of cryptococcosis. Medical mycology, 43(6), 511–516. https://doi.org/10.1080/13693780500036019
Comments 4: Quantification of growth – The manuscript does not report how many Cryptococcus colonies were obtained from each specimen (e.g. single colony versus hundreds). This information is critical in assessing the significance of the findings.
AR 4: We agree that quantitative culture can add clinical interpretability in cryptococcosis, particularly in sterile fluids (e.g., CSF, blood), where higher fungal burdens (CFU/mL) are associated with worse outcomes and where serial burden can be used to monitor treatment response. We have now made this point explicit in the Discussion, citing evidence that quantitative culture serves as a prognostic and monitoring tool, while recognising it is technically more labour-intensive and not part of most routine veterinary workflows.
Importantly, even a single colony on a sterile fluid (e.g. CSF) is clinically significant. In fungal diagnostics, any growth of Cryptococcus from a normally sterile site is considered a true positive. As one reference notes, “the reference standard for diagnosis of cryptococcosis is fungal culture” (Reagan et al., 2019), meaning that isolation of the organism confirms its presence. We have now cited this point and explained that our analysis focused on culture positivity as the endpoint. We did not have data on the number of colonies per plate, so we could not stratify by heavy vs scant growth. We have added a sentence in Discussion to explicitly mention that colony counts were not recorded and that this is a limitation. We believe this does not invalidate our results, but we thank the Reviewer for highlighting it and have made it transparent.
References
Reagan, K. L., McHardy, I., Thompson, G. R., 3rd, & Sykes, J. E. (2019). Evaluation of the clinical performance of 2 point-of-care cryptococcal antigen tests in dogs and cats. Journal of veterinary internal medicine, 33(5), 2082–2089. https://doi.org/10.1111/jvim.15599
Comments 5: Case selection – It is unclear which animals were sampled, why they were chosen, and whether they showed clinical signs consistent with cryptococcosis. Without this detail, interpretation of the data is severely limited.
AR 5: We appreciate the need to clarify our sampling frame. In response, we have expanded the Methods to describe our case selection more explicitly. All specimens came from submissions to the CEDIVET Veterinary Diagnostic Laboratory from 2013–2025. These included clinical samples (swabs, fluids, tissues) from animals presented by veterinarians across Portugal for various diagnostic reasons. We did not pre-select animals for cryptococcosis. Rather, we included all submissions to this lab for fungal culture during the study period. Each submission was accompanied by a requisition form, which we have already described.
The forms recorded the animal’s species, age, sex, breed, and importantly “clinical suspicion or observed signs (e.g. nasal discharge, neurological deficits, cutaneous lesions)”. We have added this citation and text to make it clear that only animals suspected of infection (often with relevant symptoms) were tested. In other words, this is a passive surveillance series of affected animals, not a screening of the general pet population.
To give concrete context, Table 1 now lists the types of samples yielding positive cultures. For example, among the 13 positive canine specimens, sample sources included cerebrospinal fluid (2 cases), nasal swabs (4), cutaneous swabs (4), urine (1), ear swab (1), and faeces (1). The two CSF positives correspond to dogs with neurologic signs, and the nasal swabs likely came from dogs with respiratory or nasal discharge. Similarly, the feline positives were nasal (2) or skin (3) swabs, plus one urine. We explicitly state these facts in the revised Results and Discussion to help the reader connect culture findings with clinical scenarios.
We have also added a statement that the overall denominator (1,059 animals) comprises the total diagnostic submissions during this period, drawn from multiple veterinary practices and hospitals. We clarify that this is a convenience sample (passive case series) and not a population-based survey. This context now makes clear why our findings should be interpreted as a description of diagnosed cases in this laboratory, rather than a prevalence study in the general animal population.
Comments 6: The authors may well have answers to some or all of these points, but until the methodology, case selection, and clinical context are clarified, the scientific value of this work cannot be properly assessed, and publication is premature.
AR 6: We appreciate this summary critique. We believe the extensive revisions described above now fully address the methodological and contextual concerns. In the revised manuscript, the Methods section clearly states how cases were selected and characterised (providing the details of clinical data collection), and the Results/Discussion have been amended to explain clinical context (e.g. sample types and associated symptoms). We have defended the methodological choices with appropriate citations and have also candidly acknowledged limitations (lack of antigen testing, retrospective design, incomplete metadata) in the Limitations section. With these clarifications, the study’s scientific value becomes evident: it represents a unique 13-year dataset of culture-confirmed cryptococcal isolates in Portuguese animals, integrated within a One Health perspective. We have retained all epidemiological trends (geographic and seasonal patterns) but have also adjusted the tone of conclusions to be proportionate to the data’s retrospective, passive-surveillance nature. We respectfully suggest that with the revisions, the manuscript now provides robust evidence within its scope, and that the peer review concerns have been comprehensively addressed.
We hope that the above point-by-point responses adequately address all concerns raised by the three reviewers. Throughout our revised manuscript, we have reinforced the pilot nature of the study, acknowledged its limitations (small sample size, etc.), and ensured that any claims or discussions are proportional to the data. We have defended our scientific decisions where appropriate, but always in a respectful and evidence-based manner. All changes in the manuscript are clearly indicated in the marked copy, with relevant line numbers cited in our responses above. We have also added the suggested new references.
We are grateful for the reviewers’ feedback, which has led to a significantly improved manuscript. We believe the revised version, along with this response letter, addresses all points satisfactorily. We look forward to the final editorial review and remain available to address any further questions or clarifications.

Reviewer 2 Report
I think with effort and redrafting - it will be OK - but at the moment its terrible.
I can see that a substantial amount of work has gone into this study, particularly in processing such a large number of samples. However, there are several major issues that need to be addressed before this manuscript can be considered for publication:
-
Choice of culture medium – Sabouraud’s agar with antibiotics is not truly selective. A much more appropriate choice would have been bird seed agar (Staib’s agar), where Cryptococcus colonies appear brown. This would have made identification far easier and more reliable.
-
Infection versus colonisation – It is not clear whether the isolates from animals represent true infection or simple colonisation. This distinction could readily be made using a cryptococcal antigen test (IMMY or LCAT) on serum, or through biopsies and other targeted sampling methods.
-
Background colonisation rates – Previous studies indicate that up to 25% of pound dogs and 20% of pound cats may be colonised with Cryptococcus. These animals may yield positive cultures but remain serum antigen–negative. This context is essential for interpreting your results.
-
Quantification of growth – The manuscript does not report how many Cryptococcus colonies were obtained from each specimen (e.g. single colony versus hundreds). This information is critical in assessing the significance of the findings.
-
Case selection – It is unclear which animals were sampled, why they were chosen, and whether they showed clinical signs consistent with cryptococcosis. Without this detail, interpretation of the data is severely limited.
The authors may well have answers to some or all of these points, but until the methodology, case selection, and clinical context are clarified, the scientific value of this work cannot be properly assessed, and publication is premature.
Author Response

(The authors gave the same response as above.)

Round 2
Reviewer 2 Report
I am no completely happy - but the revesion makes manythings better. So I am going to say its OK. Just OK
None